# PROBING COMPOSITIONAL FAILURES WITH CORRECTIVE PERMUTATIONS

## ABSTRACT

Modern vision models, such as Vision Transformers (ViTs), operate by decomposing images into local patches and aggregating their information for recognition. This process implicitly requires the model to not only identify the correct local features but also to correctly understand how they are spatially composed. However, this capacity for compositional reasoning is often fragile and biased. We find that in numerous misclassification cases, the model correctly attends to the right object parts, yet still yields an incorrect prediction.

This paper uncovers a surprising phenomenon: by simply permuting the arrangement of these local patches—thereby preserving local features but destroying their spatial composition—we can consistently correct these misclassifications.

We propose that this reveals the existence of **"faulty compositional information"** within the model. The original patch arrangement may trigger this flawed information, leading to failure. Our search for a corrective permutation, guided by a genetic algorithm, effectively finds an arrangement that bypasses this faulty information, forcing the model to rely on a more robust, non-compositional evidence aggregation mechanism, akin to a sophisticated bag-of-words model. Our work provides the first direct, operational tool to diagnose and understand compositional failures in vision models, highlighting a key challenge on the path toward more robust visual reasoning.

## 1 INTRODUCTION

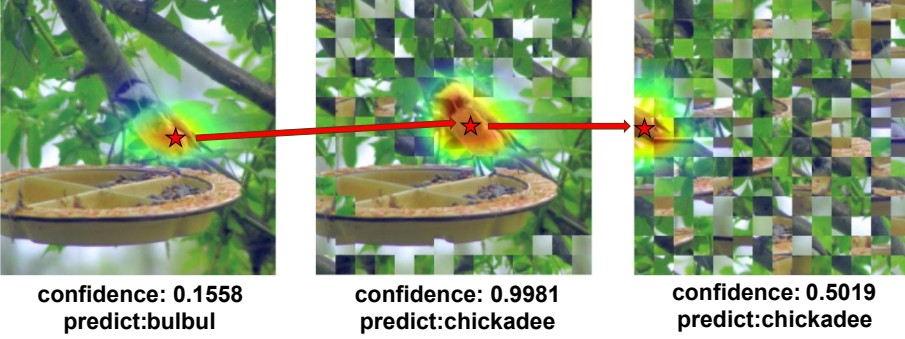

confidence: 0.1558
predict:bulbul

confidence: 0.9981
predict:chickadee

confidence: 0.5019
predict:chickadee

Figure 1: **Grad-CAM visualization before and after PPS. (Left)** The original misclassified image. The model's attention is already on the correct object (e.g., the chickadee). **(Middle)** The permuted image found by PPS, which is correctly classified. The model's attention remains focused on the chickadee's patches, despite their scrambled locations. **(Right)** The image is visually chaotic, but the model still predicts the correct label and focus on the same part of the object as the former two, which indicates that the error occurred in composing the features, not in identifying them.

Modern vision models, particularly Vision Transformers (ViTs), have achieved remarkable success by processing images as sequences of patches. This paradigm implicitly requires models to perform two fundamental tasks: first, to recognize the content of local patches (the "what"), and second, to

understand their spatial arrangement to form a coherent whole (the "how"). This latter capability, often referred to as compositional reasoning, is crucial for moving beyond simple texture recognition towards a more human-like understanding of the visual world.

However, the nature and robustness of this compositional reasoning in deep models remain poorly understood. We hypothesize that many classification errors are not failures of feature extraction, but rather failures of composition. In these cases, the model attends to the right parts, yet is misled by a brittle or spurious understanding of their spatial relationship—a classic example of shortcut learning Geirhos et al. (2020); Lapuschkin et al. (2019); Steinmann et al. (2024). How can we verify this hypothesis and systematically identify such compositional failures?

In this work, we uncover a startling and counter-intuitive phenomenon: for a significant number of images misclassified by modern vision models, we can correct the prediction by simply shuffling the order of their constituent patches. By employing a Genetic Algorithm (GA) to search over the vast space of patch permutations, we find that it is almost always possible to discover a new arrangement that guides the model to the correct answer. This happens despite the fact that the set of local features remains identical to the original image; only their spatial context is altered.

This discovery provides the first direct evidence for what we term faulty compositional information. We argue that for these failed predictions, the original patch arrangement activates a flawed reasoning path within the model, learned from statistical biases in the training data. Our Patch Permutation Search (PPS) method effectively finds an "escape route"—a permutation that breaks these spurious spatial dependencies. This forces the model to abandon its faulty compositional shortcuts and fall back on a more robust, non-compositional evidence aggregation mechanism. In essence, our method coerces the model into behaving like a sophisticated "bag-of-patches", making the correct decision based on the presence of sufficient local evidence, unburdened by misleading contextual cues.

Our contributions are threefold:

1. We are the first to identify and systematically quantify the "breaking-it-fixes-it" phenomenon, demonstrating its prevalence on major benchmarks like ImageNet-1K.

2. We propose Patch Permutation Search (PPS) as a novel, model-agnostic diagnostic tool to probe for and expose compositional failures and shortcut learning in vision models.

3. Through extensive analysis of internal representations, including heatmaps (Grad-CAM) and feature space topology (t-SNE van der Maaten & Hinton (2008), linear probing Alain & Bengio (2018)), we provide strong evidence for our "faulty compositional information" hypothesis, shedding new light on the inner workings of deep vision models.

Our work does not aim to improve model accuracy at inference time, but rather to provide a powerful new lens through which to understand and diagnose a critical failure mode. It reveals that the path to more robust visual intelligence lies not just in learning better features, but in learning to compose them correctly.

## 2 RELATED WORK

The advent of Vision Transformers (ViT) Dosovitskiy et al. (2021) and similar architectures has underscored the importance of patch order and Positional Embeddings (PE) for image understanding, with improper handling leading to significant performance decline Wu et al. (2021); Chu et al. (2023); Ren et al. (2023); Chowdhury et al. (2025); Xu et al. (2024); Jelassi et al. (2022). Beyond PE, the academic community has explored patch arrangement in various contexts. Self-supervised methods like "Jigsaw Puzzles" Chen et al. (2023) use patch order prediction as a pretext task for learning a better visual spatial representation.

Recent studies shuffle patches or apply structural perturbations to assess model robustness or to investigate biases towards texture versus shape Brendel & Bethge (2019), typically focusing on the resulting performance degradation or behavioral shifts. Naseer et al. (2021) systematically demonstrated the superiority of the ViT under various interferences, especially in the case of large-scale image occlusion, where its performance far exceeded that of CNN. Furthermore, they also found that ViT is more inclined to make decisions based on the shape of the object rather than its texture. This feature enables its off-the-shelf features to exhibit stronger generalization ability in various transfer

learning tasks. Kutscher et al. (2025) points out that the performance of VIT-type models is highly sensitive to the input order of patches. To solve this problem, they proposed the REOrder framework to learn the optimal arrangement order of image blocks for a specific task.

Surprisingly, although modern architectures such as ViT have the ability to handle spatial information through position encoding and self-attention mechanisms, a large number of studies have shown that their actual behavioral patterns often degenerate into a "bag-of-words" pattern. Yuksekgonul et al. (2022) pointed out that VLM exhibits the "bag-of-words" behavior when handling compositional tasks. And One of the core motivations for the proposal of Li et al. (2023) is to overcome the "bag-of-words" behavior of existing VLMS, which causes the model to be unable to construct words that correctly represent visual entities and their relationships. ViTs does not presuppose any prior knowledge about the structure of the image, but must learn all spatial relationships completely from the data. This enables ViT to demonstrate extremely strong flexibility and performance when it has a vast amount of training data, but it also means that the "spatial information" it learns is entirely determined by the training data. These "bag-of-words" behavior indicates that if there is some false and misleading spatial pattern in the data, the model may learn it as "wrong information".

## 3 THE CORRECTIVE PERMUTATION

In this section, we introduce an intriguing phenomenon: for many such misclassified images, merely rearranging the order of their constituent image patches can lead the same model to a correct classification, without any modification to model weights or patch content. This initial observation motivates our in-depth investigation into how and why such structural input interventions can profoundly alter model outputs.

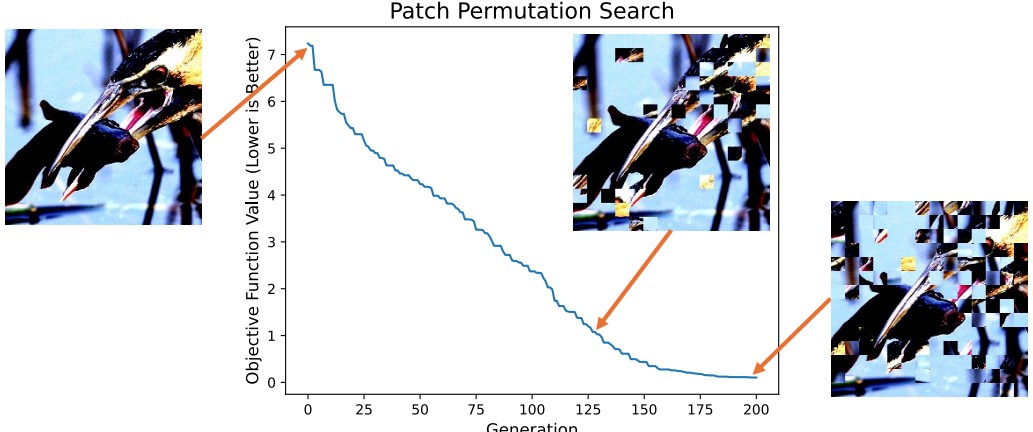

Figure 2: **Visualization of the Patch Permutation Search (PPS) process.** The central plot shows the minimization of the objective function value over 200 generations of a genetic algorithm. The image on the **left** represents the initial state (Generation 0), where patches are in their original spatial order, corresponding to a high cross-entropy value. The ground truth for this image is **"tench"**, but the model gives a prediction of **"bittern"**. As the search progresses, the patch sequence is altered. The **middle** image shows an intermediate state where the permutation is partially scrambled but already sufficient for correct classification. The final image on the **right** depicts the converged sequence, which is less human-interpretable.

### 3.1 PHENOMENON

This phenomenon is remarkably widespread. We posit that it exposes a critical vulnerability in vision models: the learning of "faulty compositional information". To probe this hypothesis, we use a Genetic Algorithm (GA) Goldberg & Holland to discover "Corrective Permutations" for misclassified images. We argue that the original, "natural" patch order of these images activates a

flawed compositional shortcut—a spurious rule based on patch co-occurrences seen during training Geirhos et al. (2020). The corrective permutation found by our GA does not create a better signal; rather, its primary function is to neutralize the misleading spatial cues. By shattering the faulty compositional structure, it compels the model to abandon its shortcut and default to a more robust, non-compositional evidence aggregation, effectively acting as a "bag-of-patches" to arrive at the correct prediction.

## 3.2 PROBLEM FORMULATION

We begin by formally defining the problem of finding a corrective patch permutation for a mis-classified image. Our work primarily focuses on patch-based vision models, such as the Vision Transformer (ViT) Dosovitskiy et al. (2021), but the formulation is generalizable to any architecture that processes images as a collection of local regions.

Let $f$ be a vision model that takes a sequence of $N$ patches as input and outputs a vector of logits over a set of classes $\mathcal{C}$. The model first transforms an input image $I \in \mathbb{R}^{H \times W \times C}$ into a spatially ordered sequence of $N$ flattened patch vectors. We denote this patch extraction and embedding process as $\mathcal{T}(\cdot)$:

$$P_{\text{orig}} = \mathcal{T}(I) = (p_1, p_2, \ldots, p_N), \tag{1}$$

where $p_i \in \mathbb{R}^D$ is the embedded vector for the $i$-th patch, typically arranged in a standard raster scan order. The model's prediction for this original sequence is given by:

$$\hat{y}_{\text{orig}} = \arg\max_{c \in \mathcal{C}} f(P_{\text{orig}})_c, \tag{2}$$

where $f(P_{\text{orig}})_c$ denotes the logit for class $c$.

We are interested in the specific set of images where the model fails. Let $y_{\text{true}} \in \mathcal{C}$ be the ground-truth label for image $I$. Our setup considers all samples for which the model's initial prediction is incorrect:

$$\hat{y}_{\text{orig}} \neq y_{\text{true}}. \tag{3}$$

A patch permutation is a bijective function $\pi : \{1, \ldots, N\} \to \{1, \ldots, N\}$ that reorders the elements of the original patch sequence. Applying a permutation $\pi$ to $P_{\text{orig}}$ yields a new sequence $P_\pi$:

$$P_\pi = \pi(P_{\text{orig}}) = (p_{\pi(1)}, p_{\pi(2)}, \ldots, p_{\pi(N)}). \tag{4}$$

Crucially, the set of local patch features $\{p_i\}_{i=1}^N$ remains identical; only their sequential order (i.e., their spatial context as interpreted by the model) is altered.

The core objective of our work is to investigate the existence and properties of a **corrective permutation**, which we denote as $\pi^*$. A corrective permutation is any permutation that, when applied to the patches of a misclassified image, successfully reverses the model's prediction to the ground-truth label. Formally, we seek to find a $\pi^* \in S_N$ such that:

$$\hat{y}_{\pi^*} = \arg\max_{c \in \mathcal{C}} f(P_{\pi^*})_c = y_{\text{true}}, \tag{5}$$

where $S_N$ is the set of all possible permutations of $N$ elements.

The central challenge is the enormous size of the search space, $|S_N| = N!$. For a standard ViT-T/16 model processing a $224 \times 224$ image, $N = 196$, which makes an exhaustive search for $\pi^*$ computationally intractable. This motivates our choice of a heuristic search method, the Genetic Algorithm, to efficiently navigate this combinatorial space.

## 3.3 PATCH PERMUTATION SEARCH VIA GENETIC ALGORITHM

Given a misclassified image $I$ and a model $f(\cdot)$, our goal is to find a patch permutation $\pi$ that corrects the prediction, i.e., $\arg\max f(\pi(I)) = y_{\text{true}}$. The search space for such permutations is astronomical ($N!$, where $N = 196$ for a standard ViT), rendering exhaustive search computationally infeasible. Furthermore, the discrete nature of the permutation operation makes the optimization problem non-differentiable, precluding gradient-based methods. To effectively navigate this vast combinatorial landscape, we propose Patch Permutation Search (PPS), a method based on a Genetic Algorithm (GA). The experimental details are in the Appendix A.1. And the process of GA is shown in Figure 2.

## 4  EMPIRICAL STUDY: PROBING COMPOSITIONAL FAILURES

In this section, we present a series of empirical studies designed to dissect the patch permutation phenomenon. Our investigation proceeds in three stages: first, we quantify the prevalence of this phenomenon across standard benchmarks (Section 4.1). Second, we analyze the characteristics of "corrective permutations" to understand what makes them effective (Section 4.2). Finally, we examine how these permutations impact the model's internal representations to reveal the mechanism behind the change in prediction (Section 4.3). Our primary models for analysis are DeiT-Tiny/16 Touvron et al. (2021) on ImageNet-1K Russakovsky et al. (2015).

### 4.1  GENERALITY AND QUANTIFICATION OF THE PHENOMENON

**Question 1: How prevalent is this phenomenon? Can any misclassified image be corrected by patch permutation?**

To answer this, we first establish whether our observation is an isolated anecdote or a general property of modern vision models. We conduct a large-scale experiment on the validation sets of ImageNet-1K. We collect all images that are misclassified by a pre-trained DeiT-T/16 model and apply our Patch Permutation Search (PPS) algorithm, implemented using a Genetic Algorithm (GA), to each of them. The fitness function for the GA is defined as the softmax probability of the ground-truth class.

Our findings are definitive: the phenomenon is remarkably general. For nearly 100% of the misclassified samples in both datasets, our PPS method is able to find at least one patch permutation that results in a correct prediction. The analysis, therefore, focuses on quantifying the computational effort required to discover such corrective permutations. As shown in Figure 3, most of the images can be corrected within a few generations of the GA, while few require a more extensive search. This suggests that for most classification errors, the necessary local features for a correct prediction are already present within the image; it is their original spatial composition that misleads the model. We can quantify the search effort required for each misclassified image $I$ by its corrective generation number, $g^*(I)$, defined as the first generation in which a corrective permutation $\pi^*$ is found:

$$g^*(I) = \min\{g \geq 0 \mid \exists \pi^* \in \text{Population}_g \text{ s.t. } \operatorname{argmax} f(\pi^*(I)) = y_{\text{true}}\} \quad (6)$$

The histogram in Figure 3 plots the empirical distribution of $g^*(I)$, where the height of a bar at generation $g$ is the count of images $|\{I \mid g^*(I) = g\}|$. The strong left skew of this distribution provides a quantitative probe for "faulty compositional information," indicating that the model's failures are often caused by fragile spatial dependencies that are easily bypassed, rather than a fundamental lack of discriminative features.

### 4.2  CHARACTERISTICS OF CORRECTIVE PERMUTATIONS

**Question 2: What are the common properties of patch permutations that correct misclassifications?**

Having established the phenomenon's prevalence, we now investigate the nature of these corrective permutations. Are they random, or do they follow specific patterns? Our core hypothesis is that these permutations succeed not by creating a new, meaningful structure, but by *destroying a specific, misleading one*. They force the model to abandon its reliance on faulty compositional shortcuts and resort to a more robust aggregation of local evidence.

**Attention remains on the salient object.** We use Grad-CAM Selvaraju et al. (2019) to visualize the model's attention before and after permutation. As illustrated in Figure 1, we consistently observe that for many original misclassified images, the model is already attending to the correct object's patches. After PPS finds a corrective permutation, the resulting Grad-CAM shows that the model's attention remains focused on the very same object parts, even though they are now spatially disjointed. This provides strong evidence that the model's failure was not one of localization, but of compositional reasoning. The permutation effectively isolates the local evidence from its misleading context. See more examples in Appendix A.3.1.

**Targeted disruption of distractors.** We uncover a more striking pattern when dealing with images containing multiple objects, especially in cases of label ambiguity or error Northcutt et al.. Consider

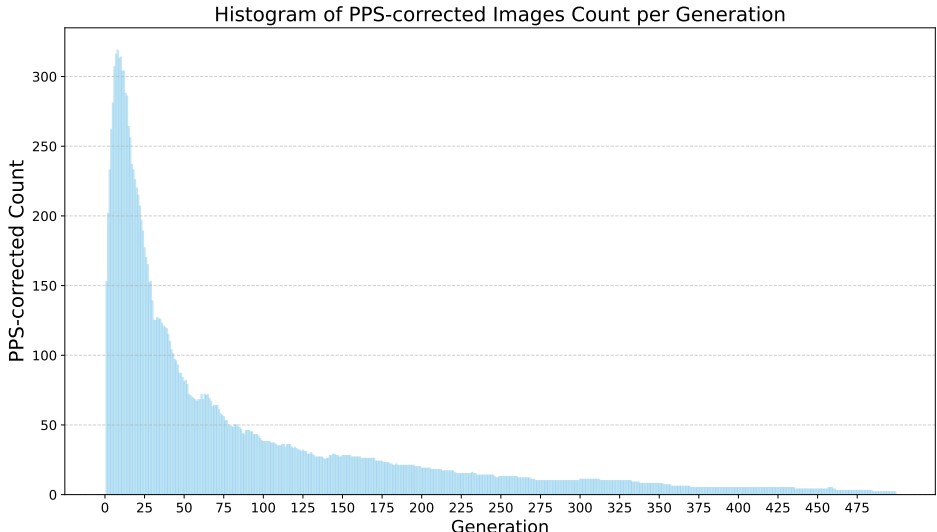

Figure 3: **Prevalency of Class-Correcting Patch Permutations.** Histogram depicting the number of successful generations—across a subset of misclassified ImageNet images runs—in which the genetic algorithm discovered a patch permutation that corrected the Vision Transformer's initial misclassification. Each bar represents how frequently a valid solution emerged at a given generation, illustrating the difficulty of the combinatorial search: valid solutions are rare in early generations and only gradually become more consistently attainable as evolution progresses.

the example in Figure 2, where an image contains a bittern holding a tench, but the ground-truth label is **"tench"**. The model initially predicts **"bittern"**. After applying PPS, the discovered permutation exhibits a remarkable property: it selectively scrambles the patches corresponding to the bird, effectively shattering its spatial structure, while leaving patches of the fish relatively intact. This suggests that PPS does not just randomly shuffle, but actively learns to dismantle the spatial information of distractor objects that contribute to the incorrect prediction.

**Initialization of GA search matters.** We also observe that the search process itself provides insights. When the GA population is initialized with the original image sequence, it tends to find solutions faster and these solutions are often visually closer to the original image. Conversely, initializing with completely random permutations also finds solutions, but they are typically more chaotic and take longer to converge, just like the right image shown in Figure 1. It looks as if the model is forced to process the images with the behavior of "bag of words". This suggests the existence of multiple "solution basins" in the permutation space—some involving minor perturbations to break a specific shortcut, and others requiring a complete tear-down of spatial structure.

### 4.3 IMPACT ON INTERNAL REPRESENTATIONS

**Question 3: How does patch order mechanistically alter the model's internal representations to change the prediction?**

Finally, we move from external observations to the model's internals. We analyze the final [CLS] token representation, which aggregates information from all patches and is fed to the classification head. We aim to understand how PPS alters this representation to make it "correctable."

**Qualitative analysis via t-SNE.** We use t-SNE to visualize the [CLS] token embeddings of images in different states: (1) original misclassified, (2) randomly permuted, and (3) PPS-corrected. As shown in Figure 4, the representations for PPS-corrected images form distinct clusters. Notably, solutions found from an original-image initialization often lie in the feature space as "neighbors" to the original misclassified point, suggesting a subtle refinement of the representation. In contrast, solutions from random initializations can be located in a completely different region of the space, in-

dicating that PPS can discover entirely new, effective reasoning paths. More comprehensive analysis can be found in Appendix A.3.2.

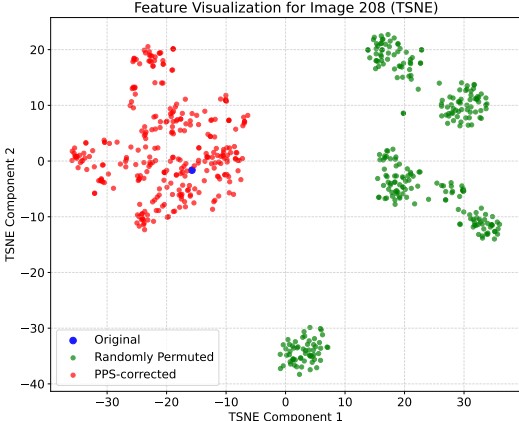

(a) t-SNE embedding space for the [CLS] tokens

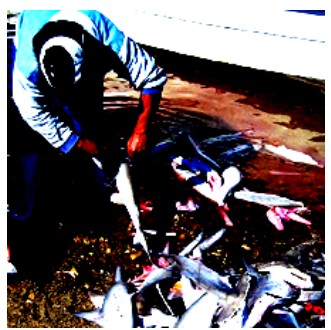 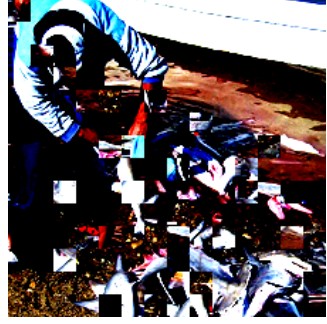 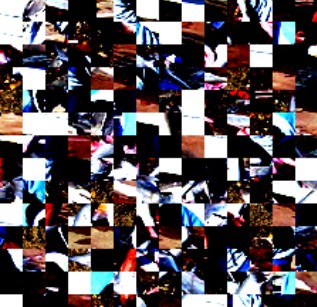

(b) Original misclassified image: Person handling fish, representing the blue point in (a)

(c) PPS corrected image : representing an instance of the red points in (a)

(d) Randomly permutated image: representing an instance of green points in (a)

Figure 4: Analysis of PPS correction for a misclassified image. (a) t-SNE reveals distinct clusters of original misclassified embeddings (blue), randomly permuted images (green), and PPS-corrected embeddings (red). (b) Original misclassified image. (c) PPS correction with original-sequence initialization preserves spatial coherence. (d) Random initialization leads to fragmented patch arrangements, demonstrating the critical role of initialization strategy in effective correction.

**Quantitative analysis via Linear Probing.** To quantify the quality of these representations, we perform a linear probing experiment. We freeze the DeiT backbone and train a linear classifier on the [CLS] token embeddings. We test the separability of features from three groups of images, all of which are originally misclassified (their baseline accuracy is 0%). The results, summarized in Table 1, are striking. The representations of randomly permuted images show a modest increase in linear separability (about 30% accuracy). This suggests that simply breaking the original structure can sometimes resolve conflicting signals. However, the representations generated by our PPS-corrected images are almost perfectly linearly separable, achieving nearly 99% accuracy.

This result provides the strongest evidence for our hypothesis. PPS is not merely finding a brittle decision boundary crossing; it is actively reshaping the internal feature representation into one that is far more semantically coherent and robustly separable. The model is guided to a region in the latent space where the evidence for the correct class is overwhelming and unambiguous, a state it could not reach with the original, misleading spatial arrangement.

Table 1: **Linear probing accuracy on DeiT-T/16 [CLS] token representations.** We train a linear classifier on features from images that were originally misclassified by the full model. The dramatic increase in accuracy for PPS-corrected images shows they are represented in a much more linearly separable way.

| Image Representation Source | Linear Probe Accuracy |
| --- | --- |
| Original (Misclassified) Images | 0% (by definition) |
| Randomly Permuted Images | 30.05% |
| **PPS-Corrected Images** | **98.45%** |

## 5 Discussion and Implications

Our empirical results demonstrate that for a vast majority of misclassified images, a "correct" prediction is hiding in plain sight, accessible by merely re-arranging image patches. This finding has profound implications for how we understand and evaluate modern vision models.

### 5.1 A General Phenomenon Beyond Vision Transformers

**Is this a ViT-specific artifact?** A natural question is whether our findings are unique to the Vision Transformer architecture, perhaps stemming from a vulnerability in its position embedding or self-attention mechanisms. To investigate this, we replicated our experiments on a standard ResNet-based CNN architecture He et al. (2015). We partitioned the input image into a grid of patches and treated them as a sequence, feeding them to the CNN. We observed the exact same phenomenon: for misclassified images, a Genetic Algorithm could find permutations of these patches that corrected the model's prediction.

This crucial result indicates that the issue is not tied to a specific architectural component like position embeddings. Rather, it points to a more fundamental weakness in how deep neural networks learn to aggregate spatial evidence. Both convolutional and self-attention layers are designed to build hierarchies of features based on local correlations. Our work suggests that in doing so, they often learn brittle, spurious compositional rules that constitute a form of spatial shortcut learning.

### 5.2 Patch Permutation Search as a Diagnostic Tool

We do not propose PPS as a method for improving inference-time accuracy due to its computational cost. Instead, its primary value lies in its role as a powerful **diagnostic tool** to probe the compositional reasoning of any patch-based vision model. It opens up new avenues for model and data analysis:

**Model Auditing and Robustness Evaluation.** PPS can be used to quantify a model's reliance on fragile spatial cues. Given two models with similar overall accuracy, the model for which PPS can correct a higher fraction of errors is likely the one relying more heavily on non-robust compositional shortcuts. This provides a new, targeted metric for evaluating model robustness that complements traditional approaches like adversarial attacks or out-of-distribution testing.

**Dataset Analysis and Anomaly Detection.** Our "bird-and-fish" case study (Figure 2) highlights a fascinating application: using PPS to debug datasets. The targeted scrambling of the "bird" (the distractor) to achieve the correct "fish" label suggests that PPS can automatically identify and isolate parts of an image that are in conflict with the ground-truth label. This could be used to flag images with potential multi-label situations, incorrect labels, or other dataset artifacts that might confuse a model during training.

### 5.3 LIMITATIONS AND FUTURE WORK

Our work opens several exciting directions for future research. While PPS is an effective probe, it is currently a post-hoc analysis tool. A key future direction is to leverage these insights to build inherently more robust models. This could take several forms:

- **Adversarial Training:** One could formulate a training objective where the model is encouraged to be invariant to permutations that do not destroy the core object, effectively performing a form of "compositional adversarial training."
- **Regularization:** Can we design a regularization term that explicitly penalizes over-reliance on specific, rigid spatial configurations?
- **Architectural Innovations:** Our findings motivate the development of new architectures that better disentangle the representation of "what" (local content) from "how" (spatial composition), potentially through modular or neuro-symbolic approaches.

Furthermore, extending this probe to other vision tasks like object detection or segmentation could yield valuable insights into their respective failure modes.

## 6 CONCLUSION

In this paper, we introduced and systematically studied a counter-intuitive yet widespread phenomenon in modern vision models: the ability to correct a misclassification by simply shuffling the image patches. We have shown that this is not an anomaly but a systematic behavior that reveals a critical vulnerability we term faulty compositional information. Our findings suggest that many classification errors are not due to a failure in recognizing local features, but a failure in correctly composing them, often because the model has latched onto brittle spatial shortcuts.

We proposed the Patch Permutation Search (PPS) not as a performance-enhancing trick, but as the first direct, operational tool to diagnose these compositional failures. By analyzing the effects of PPS through heatmap visualizations and internal representation analysis, we provided strong evidence that it works by breaking flawed spatial dependencies and forcing the model into a more robust, non-compositional mode of evidence aggregation. Ultimately, our work provides a new lens for understanding the complex inner workings of deep vision systems and underscores a key challenge on the path toward more robust and generalizable visual intelligence: learning not just what to see, but how to see it.

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

# A APPENDIX

## A.1 GENETIC ALGORITHM SETUP

A GA is a population-based metaheuristic inspired by natural selection, well-suited for this task. We define the core components of our GA as follows:

**Chromosome and Population** Each individual solution in the population, or *chromosome*, is represented as a permutation $\pi$ of the patch indices, i.e., a sequence of unique integers from $\{0, 1, \ldots, N - 1\}$. A population consists of a fixed number of such chromosomes.

**Fitness Function** The objective is to find a permutation that leads to a correct and confident prediction. We define the fitness of a chromosome $\pi$ as the cross-entropy loss between the model's prediction for the permuted image $\pi(I)$ and the ground-truth label $y_{\text{true}}$. The GA's objective is to **minimize** this fitness score:

$$\text{Fitness}(\pi) = L_{\text{CE}}(f(\pi(I)), y_{\text{true}}) \tag{7}$$

A lower fitness value indicates that the corresponding patch permutation brings the model's output closer to the true label.

**Genetic Operators** The evolution process is driven by selection, crossover, and mutation. In each generation, a portion of the population with the best fitness scores is selected as parents. New offspring are generated using:

- **Crossover:** We employ Partially-Mapped Crossover (PMX), a standard operator for permutation-based chromosomes that ensures the offspring are valid permutations.
- **Mutation:** A swap mutation is applied, which randomly selects two indices in the chromosome and swaps their values.

Additionally, we incorporate elitism, where a small fraction of the best-performing individuals from the current generation are directly carried over to the next, ensuring that the best-found solution is never lost. The specific hyperparameters used for our GA are detailed in Table 2.

**Initialization Strategies** The initial population critically influences the search trajectory. We explore two distinct strategies to understand its impact:

- **Identity Initialization:** The entire initial population is seeded with the original patch sequence, $\pi_{\text{orig}} = (0, 1, \ldots, N - 1)$. This strategy initiates the search from the vicinity of the original image configuration, exploring permutations that are structurally "close" to the original.
- **Random Initialization:** The population is initialized with completely random permutations. This encourages a broader, more exploratory search across the entire permutation space, without any bias from the original spatial layout.

As we will demonstrate in our experiments, these two strategies converge to solutions with remarkably different characteristics. Identity-initialized searches often yield minimally perturbed, visually recognizable images, whereas random-initialized searches converge to highly chaotic and indecipherable patch arrangements.

**Early Stopping Criterion** A key aspect of our experimental design is the inclusion of an early stopping mechanism. While we set a maximum of 500 generations for the search, the process for any given image is terminated as soon as a permutation $\pi$ is found for which the model's top-1 prediction, $\arg\max f(\pi(I))$, matches the ground-truth label $y_{true}$. This allows us to not only find a corrective permutation but also to measure the *search efficiency*, i.e., the number of generations required to fix the prediction. As shown in our results (Figure 3), this early stopping is triggered for nearly all samples, often well before the maximum generation limit is reached, highlighting the widespread accessibility of these corrective solutions in the search space.

Table 2: Hyperparameters for the Genetic Algorithm used in Patch Permutation Search (PPS).

| Parameter | Value |
|---|---|
| Max Iterations | 500 |
| Population Size | 500 |
| Parents Portion | 0.3 |
| Crossover Probability | 0.8 |
| Crossover Type | PMX |
| Mutation Probability | 0.8 |
| Mutation Type | Swap |
| Elitism Ratio | 0.01 |

## A.2 LINEAR PROBING SETUP

- **Datasets:** Our primary experimental platform is ImageNet-1K. Images ($224 \times 224$) are divided into $N = 196$ non-overlapping $16 \times 16$ patches.

- **Models:** We evaluate two distinct architectures: Vision Transformer (DeiT-T) and a Convolutional Neural Network (ResNet-18). All models are pre-trained on ImageNet-1K.

  - **Training Data:** Linear probes are trained using features before the linear classification head generated from three types of sequences: Features from the original image patch sequence, Features from randomly shuffled patch sequences and features from image patches reordered by corrective permutation.

  - **Hyperparameters:** Training involved 60 epochs, a learning rate of $1 \times 10^{-3}$, the AdamW optimizer, and a batch size of 128. The dataset was split into an 80% training set and a 20% test set.

We not only apply linear probing on ViT, but also on ResNet:

Table 3: **Linear probing accuracy on ResNet-18 last layer token representations.** We train a linear classifier on features from images that were originally misclassified by the full model. The dramatic increase in accuracy for PPS-corrected images shows they are represented in a much more linearly separable way.

| Image Representation Source | Linear Probe Accuracy |
|---|---|
| Original (Misclassified) Images | 0% (by definition) |
| Randomly Permuted Images | 22.37% |
| **PPS-Corrected Images** | **78.29%** |

From this, it can be seen that ResNet did not achieve the same high accuracy as ViT Table 1 on linear probing. In contrast, the design philosophy of ResNet (and all CNNs) is built upon strong, fixed spatial inductive biases. The essence of the convolution operation is to process spatially adjacent local regions, gradually expanding the receptive field through layered stacking. Its entire workflow heavily relies on the spatial continuity and locality of input features. When PPS feeds a spatially fragmented sequence to ResNet, it directly violates the model's most fundamental working assumption. The convolutional kernels cannot effectively integrate information by crossing non-adjacent

blocks, even if these blocks are semantically related. Therefore, although the 78.29% accuracy suggests that the model can still benefit from semantically clustered low-level features (such as color and texture), its inherent spatial dependency limits its ability to learn high-level global features from this "disordered" input.

### A.3 MORE ON VISUALIZATION

This section provides additional visualizations to supplement the empirical findings presented in the main paper. These examples further illustrate the key phenomena of attention stability, internal representation shifts, and the evolutionary search process.

#### A.3.1 GRAD-CAM

To further illustrate the findings from Section 4.2, Figure 5 provides additional examples of Grad-CAM visualizations. These examples reinforce the observation that the model's attention is often correctly localized on the salient object even when the final prediction is wrong. In each pair, the top row shows the original misclassified image, and the bottom row shows a PPS-corrected version. Despite the radical scrambling of patches, the model's attention (heatmaps) remains consistently focused on the same set of patches belonging to the target object. This provides strong evidence that the initial error was compositional—a failure to interpret the spatial arrangement of correctly identified features—rather than a failure of feature localization itself.

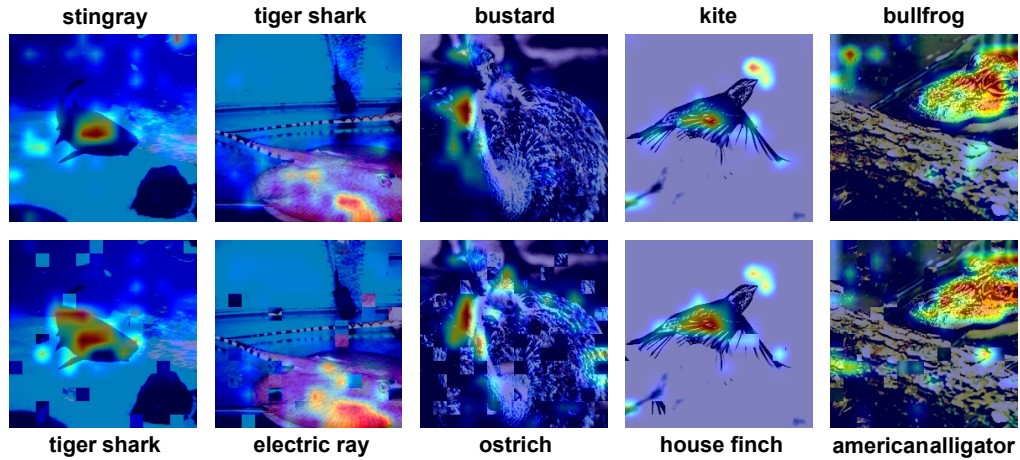

Figure 5: **Additional Grad-CAM Examples.** The top row of each pair shows the original misclassified image and the model's prediction. The bottom row shows the image after a corrective permutation has been applied, leading to the correct prediction (the ground truth). The model's attention remains fixed on the key object's patches throughout.

#### A.3.2 MORE VISUALIZATION ON [CLS] TOKEN

We expand upon the analysis of internal representations presented in Section 4.3. Figure 6 presents additional t-SNE visualizations and corresponding cosine similarity distributions for the [CLS] token embeddings of several misclassified images.

The left column (t-SNE plots) consistently demonstrates the geometric separation in the feature space: the original misclassified embedding (blue) is distinct from the cloud of random permutations (green), while the PPS-corrected embeddings (red) form a tight, separate cluster. This shows that corrective permutations are not random but belong to a specific, structured region of the permutation space.

The right column quantifies the relationships within the feature space. It plots the distribution of cosine similarities calculated between the [CLS] token of the original misclassified image and two other sets of embeddings: those from PPS-corrected images (red) and those from randomly per-

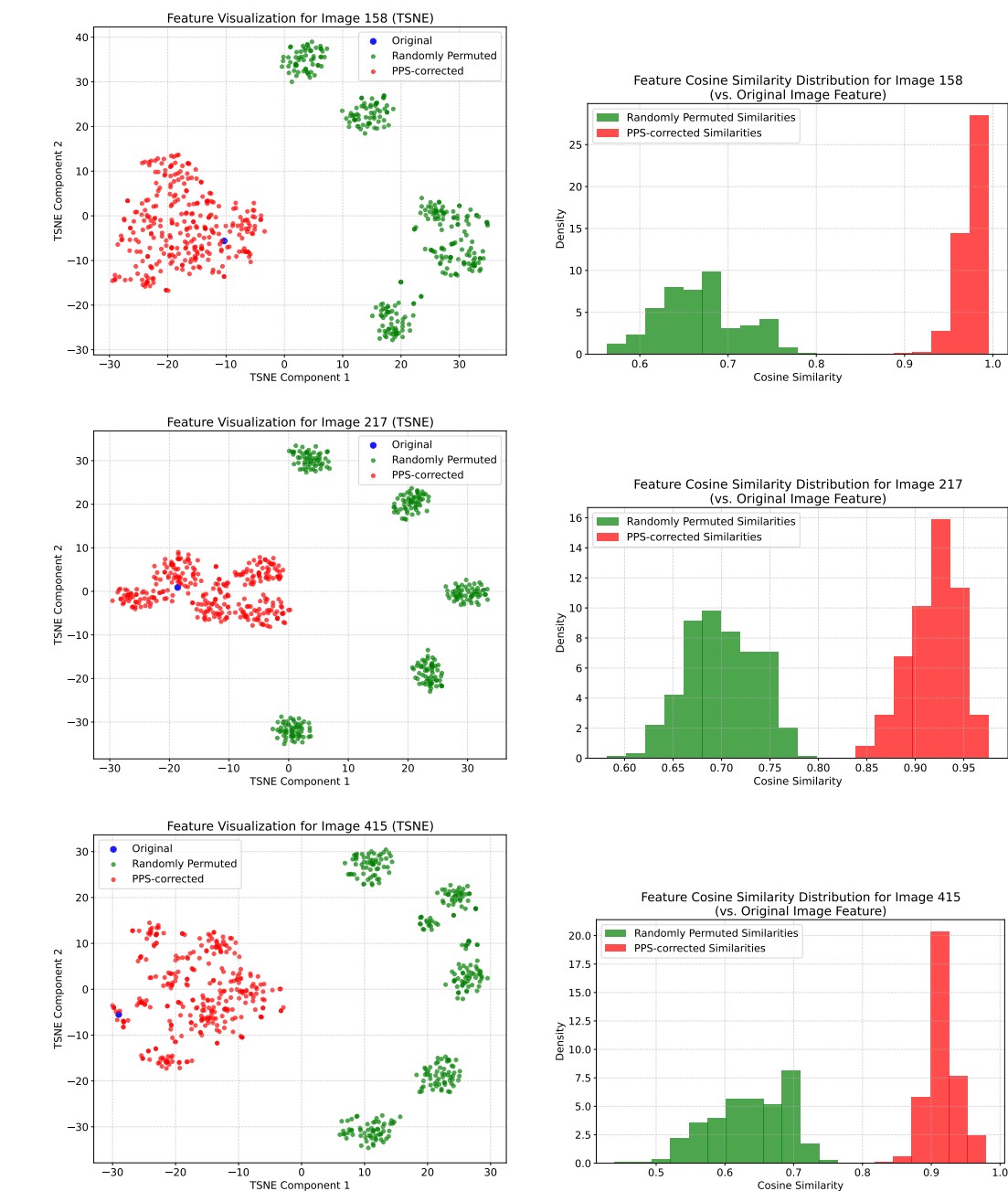

Figure 6: **Additional t-SNE and Cosine Similarity Visualizations of [CLS] Token Embeddings.** Each row corresponds to a single misclassified image. (Left) t-SNE plots showing original (blue star), randomly permuted (green cloud), and PPS-corrected (red cloud) embeddings. (Right) Histograms show the cosine similarity of PPS-corrected (red) and randomly permuted (green) embeddings relative to the original embedding. The PPS-corrected solutions are significantly more similar to the original than random permutations are.

muted images (green). Let $v_{\text{orig}}$, $v_{\text{pps}}$, and $v_{\text{rand}}$ be the [CLS] token representations for the original, a PPS-corrected, and a randomly permuted image, respectively. The histogram visualizes two distributions of cosine similarities: the similarity between PPS-corrected and original embeddings, $\{\cos(v_{\text{pps}}, v_{\text{orig}})\}$, shown in red, and the similarity between randomly permuted and original embeddings, $\{\cos(v_{\text{rand}}, v_{\text{orig}})\}$, shown in green.

The consistent rightward shift of the red distribution indicates that, on average, the representations of corrective permutations are much more similar to the original representation than those of random permutations. The observation can be expressed as:

$$\mathbb{E}_{\pi_{\text{pps}} \sim \text{PPS}}[\cos(v_{\text{pps}}, v_{\text{orig}})] \gg \mathbb{E}_{\pi_{\text{rand}} \sim \text{Uniform}}[\cos(v_{\text{rand}}, v_{\text{orig}})]$$

This implies that while random permutations push the image's representation to a distant, unrelated region of the latent space, PPS finds corrective solutions whose representations remain relatively close to the original. It suggests that PPS is not finding an arbitrary path to a correct classification but is instead performing a targeted "nudge" on the feature vector. It preserves much of the core feature information from the original image while subtly altering it just enough to escape the pull of the incorrect class and move into the basin of attraction for the correct one.

### A.3.3 PATCH PERMUTATION SEARCH PATH

To visualize the optimization process of the Patch Permutation Search, Figure 7 tracks the evolution of the model's heatmap over several generations of the genetic algorithm. Each row corresponds to a different misclassified image, progressing from an early generation (left) towards a converged, corrective permutation (right).

The key insight from these trajectories is that the model's focus, as indicated by Grad-CAM, is established on the correct object early in the search and remains remarkably stable throughout the optimization. The GA is not helping the model find the object; the model has already found it. Instead, the GA's role is to discover a patch arrangement that dismantles the misleading spatial context, allowing the already-localized features to be aggregated in a way that leads to a correct classification. This dynamic view of the search process further reinforces our central hypothesis about faulty compositional information.

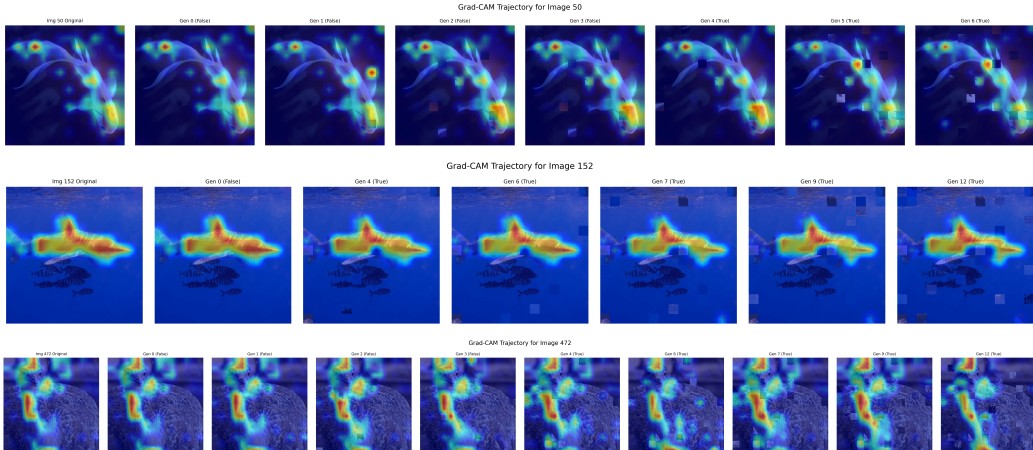

Figure 7: **Evolution of Grad-CAM Attention During Patch Permutation Search.** Each row tracks the attention map for a single image across different generations of the GA (progressing left to right). The heatmaps consistently highlight the core object, demonstrating that the model's focus on discriminative regions is preserved even as the spatial structure is radically altered by the search algorithm.

