# OpenReview forum: "Probing Compositional Failures with Corrective Permutations"
_ICLR.cc/2026/Conference — ICLR 2026 Conference Withdrawn Submission_

### Official Review · Reviewer_X11E · 2025-11-01

**Soundness:** 2
**Presentation:** 3
**Contribution:** 2
**Rating:** 4
**Confidence:** 3

**Summary:**

The paper introduces and investigates a counter-intuitive phenomenon in modern vision models like ViTs where, for a significant number of images that are misclassified by the model, the model can be guided to the correct prediction by simply shuffling or permuting the order of the image's patches. The authors hypothesize that these errors are not failures of feature extraction, since the patches that the model attends are generally consistent across permutations, but are instead failures of compositional reasoning. It then introduces Path Permutation Search (PPS), a genetic algorithm that searches through the space of all permutations of patches to find the "corrective permutation" that maximizes the probability that the image is classified correctly. They then perform an empirical study on ImageNet-1k using the DeiT-T/16 model as the classifier. They find that PPS discovers a permutation that results in the correct classification for nearly all misclassified samples. They also perform a qualitative analysis with t-SNE to visualize the classification token embeddings of the original misclassified image, the PPS corrected image, and a random permutation of patches, as well as a quantitative analysis via linear probing that showed that the representations found by PPS were almost perfectly linearly separable.

**Strengths:**

- The discovery that patch permutation can correct misclassifications is novel and counter-intuitive.
- The combined Grad-CAM, t-SNE, and linear probing analyses provides evidence from multiple perspectives, which strengthens empirical rigor.
- The paper is clear and well-written, and the authors are transparent about the method's limitations, in particular, acknowledging PPS is diagnostic and not practical for inference.
- The work reveals that spatial composition in vision models is surprisingly brittle and manipulable, which has significant implications for robustness and reliability.

**Weaknesses:**

- Most critically, the work only tests steering toward ground truth labels, and the search process explicitly optimizes on classification accuracy according to the ground truth label. Can PPS equally easily find permutations that steer misclassifications toward other wrong labels (e.g. steering the bulbul classification from Figure 1 toward another bird species instead of the ground truth chickadee)? Can PPS find permutations that cause initially correct classifications to fail? If yes, this would show general decision boundary manipulability rather than revealing specific compositional failures.
- The baselines for the linear probing analysis are unfair given that PPS-corrected images (which keep searching until the correct classification is reached) are compared only against the original misclassified image and randomly permuted patches. A more fair baseline would compare the PPS-corrected images to natural images that are correctly classified and randomly permuted patches that happen to be classified correctly.
- Many objects genuinely require spatial composition for recognition. By destroying composition to "fix" predictions, the method may be discarding necessary information and forcing texture-based shortcuts, not revealing compositional failures. For example, the difference between a bike and a collection of tires and metal is the compositional of the two tires and a specific frame in a way that is structurally meaningful. The claim that the "bag of words" representation is generally more robust than ones that preserve less localized structure is not well supported. The paper would benefit from more nuanced discussion regarding the assumed superiority of the "bag of words" representation.

**Questions:**

Please address the questions and concerns mentioned in the weaknesses. Additionally:
- What is the computational cost and time needed per image for PPS?
- In Figure 4a, do the green points represent randomly permuted patches or PPS-corrected embeddings initialized from randomly permuted patches? If they are just randomly permuted patches, then the observation of clustering of PPS-corrected embeddings around the original image seems like it would just be a result of initializing PPS on the original image. It would be interesting to see if there is any clustering of PPS-corrected embeddings initialized from randomly permuted patches.

---

### Official Review · Reviewer_itpf · 2025-11-01

**Soundness:** 2
**Presentation:** 2
**Contribution:** 3
**Rating:** 2
**Confidence:** 4

**Summary:**

The paper proposes a diagnostic method termed Patch Permutation Search (PPS). For images that a model misclassifies, the input is partitioned into local patches, and a genetic algorithm is used to explore the vast permutation space to identify a corrective permutation. The authors’ key observation is that, across numerous misclassified examples, keeping the set of local features fixed while perturbing only their spatial composition often flips the prediction from incorrect to correct. They interpret this as evidence of faulty compositional information: the model attends to the appropriate local cues, but takes a shortcut in how those cues are composed—i.e., it relies on spurious composition.

**Strengths:**

- The paper operationalizes the phenomenon that “models correctly identify local evidence yet still misclassify” as a permutation search problem. Using Grad-CAM visualizations, it shows that the attended regions remain largely unchanged before and after correction, highlighting that the error lies in how cues are composed rather than what is perceived.
- Beyond qualitative visualizations, the study reports t-SNE separability and linear-probe results, supporting the claim that corrective permutations push representations into more linearly separable regions.
- Similar trends are observed on CNNs (e.g., ResNet-18), suggesting that the issue is not a quirk of a specific ViT component (such as positional encoding) but reflects a broader vulnerability in how deep models aggregate spatial evidence.

**Weaknesses:**

- PPS is framed as a diagnostic tool, but the paper does not convert its signals into actionable modeling or data interventions (e.g., regularizers, architectural changes, or cleaning pipelines) or verify reduced reliance on corrective permutations. Suggestion: Implement a minimal variant of compositional adversarial training/regularization/architectural decoupling and report whether PPS-correctability on misclassified samples decreases.
- The method casts permutation search as a genetic algorithm, while patch shuffling/reordering is well studied; the contribution is chiefly a diagnostic assembly rather than a new algorithm.
- Results center on ImageNet-1K with DeiT-T (plus ResNet-18). To support generality, add datasets (CIFAR-100, iNaturalist, WILDS), larger/backbone-diverse models (ViT-B/L, ConvNeXt, Swin, SAM), and downstream tasks (detection, segmentation, multi-label).
- Core method and implementation details sit in the appendix, while motivation/case analyses dominate the main text. Suggestion: move the optimization objective, operators/initialization, early stopping, and complexity into Methods, and include a minimal reproducible hyperparameter table.

**Questions:**

- Corrective permutations can bypass faulty compositional information, but what practical value does this provide—does PPS inform concrete model/training design, or does it merely reveal a limitation without a remedy? Specify an actionable pathway (e.g., regularization, architecture, data cleaning) and verify whether reliance on corrective permutations decreases after such interventions.
- The method appears to be a straightforward application of a genetic algorithm to permutation search. What necessitates GA over stronger baselines (multiple random shuffles, greedy/heuristic reorders, REOrder-style task-optimal orderings)? Provide quantitative comparisons (success rate, iterations, function evaluations) and ablations over operators/initialization.
- If the aim is improving target sensitivity, why are comparisons with established methods absent? Report relevant baselines and contextualize PPS accordingly.

---

### Official Review · Reviewer_G4Cs · 2025-11-01

**Soundness:** 2
**Presentation:** 2
**Contribution:** 2
**Rating:** 4
**Confidence:** 4

**Summary:**

This paper proposes Patch Permutation Search (PPS) as a diagnostic tool to reveal compositional failures in vision models. By applying a genetic algorithm to search for corrective permutations of image patches, the method shows that many misclassified samples in modern architectures such as ViT and ResNet can be correctly reclassified without modifying model weights or local features.

**Strengths:**

1. The paper introduces a novel and operational diagnostic method for probing shortcut learning and compositional failures, addressing a gap between theoretical analyses and empirical verification.
2. Cross-architecture validation on ViT and CNN backbones strengthens the generality of the findings and links the observed phenomenon to broader representation behaviors like “bag-of-patches” degeneration.

**Weaknesses:**

1. The claim that almost all misclassifications can be corrected by PPS requires stronger statistical grounding, for instance, clearer reporting of search budgets, termination conditions, and sample stratification.
2. The diagnostic signal may be confounded with dataset noise or multi-label ambiguity, as PPS might “correct” mislabeled or ambiguous samples rather than expose true compositional failures.
3. The paper lacks comparisons with differentiable or learning-based permutation methods, such as Gumbel-Sinkhorn relaxation or REOrder [1], which could contextualize PPS within broader optimization paradigms.
4. Experiments are limited to small-scale models and classification tasks, leaving open questions for detection or segmentation applications.

[1] Declan et al. REOrdering Patches Improves Vision Models.

**Questions:**

1. How does the trade-off between correction rate and search cost behave? For a fixed compute budget, what is the recall curve of PPS under varying population sizes and mutation rates?
2. Can the mechanism be formalized under permutation equivariance or invariance theory, which specific positional encodings or attention layers are most responsible for faulty compositions?
3. How could PPS integrate with training-stage mitigation, e.g., using discovered “bad compositions” as adversarial samples or regularization signals, similar to REOrder’s patch order learning?
4. Could controlled experiments on synthetic data with known label noise or compositional rules disentangle whether PPS corrects due to noise avoidance or genuine reasoning repair?

---

### Official Review · Reviewer_Naem · 2025-11-01

**Soundness:** 3
**Presentation:** 3
**Contribution:** 2
**Rating:** 4
**Confidence:** 3

**Summary:**

This paper presents a method that identifies permutations of image patches that can correct misclassifications in vision models. The authors argue that such corrections reveal a failure mode rooted in faulty compositional reasoning. While the phenomenon is interesting and the empirical analysis is thorough, the method is limited to per-instance corrections, does not generalize, and largely reiterates insights that can already be obtained through attention-based interpretability tools. Its practical and theoretical value remains narrow, making it insufficient for acceptance in its current form.

**Strengths:**

● The paper identifies an unusual and counter-intuitive observation. This sheds light on spatial biases and compositional fragility in vision models.

● The authors support their findings with diverse experiments, including Grad-CAM visualizations, t-SNE of feature embeddings, and linear probing, providing a multi-angle look at how corrective permutations affect model behavior.

**Weaknesses:**

● PPS only works on a per-image basis. Each corrective permutation is found via a costly optimization for a single misclassified sample, with no reusable insight or impact on model behavior. It does not generalize to other samples nor lead to systematic model improvements.

● The main insight—that models often attend to the correct object but misclassify due to compositional errors—is already obtainable via existing tools like Grad-CAM. PPS essentially reconfirms this known failure mode through an expensive patch-level manipulation, offering limited incremental insight.

● Corrective permutations often produce highly disordered images that are not human-interpretable. This weakens the claim that the model “understands” the correct object post-permutation. Instead, it highlights the model’s fragility and tendency to rely on non-semantic statistical cues.

● The paper provides no theoretical framework explaining why corrective permutations exist or which types of errors they can resolve. This limits the contribution to an empirical curiosity rather than a foundation for future compositional reasoning research.

● Experiments are restricted to image classification on ImageNet using ViT and ResNet. No results are presented on detection, segmentation, or larg

**Questions:**

● Since each corrective permutation is optimized for a single image, do you see any potential for generalization beyond isolated cases?

● Given that Grad-CAM already highlights attention misalignment, what does PPS reveal that attention maps do not?

● Have you compared PPS to simpler alternatives, such as targeted occlusion or patch masking, in terms of correcting misclassifications?

● The method appears tightly bound to classification tasks. Is there any evidence it would extend to structured prediction tasks like detection or segmentation?

● Did you observe any interpretable patterns in the corrective permutations, or are the solutions essentially arbitrary per image?

---

### Note · Authors · 2025-11-17

**Comment:**

We would like to thank reviewers for the constructive feedback. We will incorporate the feedback in a newer version. Thank you.

**Withdrawal Confirmation:**

I have read and agree with the venue's withdrawal policy on behalf of myself and my co-authors.